# Survival Analysis to Predict How Color Influences the Shelf Life of Strawberry Leather

**DOI:** 10.3390/foods11020218

**Published:** 2022-01-13

**Authors:** Raquel da Silva Simão, Jaqueline Oliveira de Moraes, Julia Beims Lopes, Ana Caroline Cichella Frabetti, Bruno Augusto Mattar Carciofi, João Borges Laurindo

**Affiliations:** Department of Chemical and Food Engineering, Federal University of Santa Catarina, EQA/CTC/UFSC, Florianópolis 88040-970, Brazil; raquelsimao42@gmail.com (R.d.S.S.); jaquelinemoraes111@gmail.com (J.O.d.M.); juliabeims@gmail.com (J.B.L.); anacfrabetti@gmail.com (A.C.C.F.); bruno.carciofi@ufsc.br (B.A.M.C.)

**Keywords:** fruit leather, current-status survival analysis, color, anthocyanin, light, relative humidity

## Abstract

Color change of fruit-based products during storage is an important quality parameter to determine their shelf life. In this study, a combination of relative humidity (RH) and illumination was evaluated on the stability of strawberry leathers. Samples were conditioned at 25 °C, in chambers with RH of 22.5% and 52.3% and under two levels of illumination (no illumination and with a light-emitting diode (LED) illumination at 1010 lx). Samples were analyzed during storage by instrumental color measurements, total anthocyanin content, and consumers’ acceptance/rejection of the product color. Current-status survival analysis was performed to estimate the sensory-based shelf-life of the strawberry leather. The chromatic parameters (*a** and Δ*E** values) and anthocyanin content changed with increasing storage time and RH, fitting a first-order fractional conversion model. Samples conditioned at the higher RH showed a higher reduction of *a** values and anthocyanins losses when stored under LED illumination than those without illumination. The increase of RH resulted in a faster increase of the consumer rejection probability and a shorter shelf life of the strawberry leather. For 50% of consumers’ rejection, the sensory shelf life of the strawberry leather equilibrated at 22.5% RH was estimated as at least 54 days, while it was reduced to approximately 2 days at 52.3% RH. The red chromatic parameter (*a** value) strongly correlated to the percentage of consumer rejection in all storage conditions, suggesting that this analytical parameter can be useful as a predictor of strawberry leather’s shelf life. Therefore, the results of this study show the applicability of an approach that integrates instrumental and sensory data to acquire faster information on color changes during the storage of strawberry leather and product shelf-life prediction.

## 1. Introduction

Fruit leather, also known as fruit bar or restructured fruit, is a product in the form of a pliable strip or sheet obtained by thin-layer drying of fruit pulp, with or without the incorporation of additives. It is an alternative to preserving, adding value and diversifying consumption options of various fruits. This product is a typical healthy snack in North America, Africa, and Asia. The relatively low moisture content, intermediate water activity, and low pH value of fruit leather result in microbiological stability of the product for up to 6 months [1]. The shelf life of fruit leathers and many food products is stated by altering their sensory aspect before their microbiological safety is compromised [2]. According to Buvé et al. [3], the color change during storage is the main parameter that predicts the shelf life of fruit-based products.

Food color plays an essential role in influencing the sensory and hedonic expectations the consumer has regarding foods during the search for, purchasing, and consumption of food [4]. Since color is a visual property, changing it can cause product rejection, even on the market shelves. The attractive red color of strawberry-based products, such as strawberry leathers, can be easily changed during storage and replaced by a dull brownish color [5]. This color loss is attributed to the degradation of anthocyanins, which are compounds responsible for the red color of strawberries, as well as the enzymatic and non-enzymatic browning reactions [3]. The color stability depends on many factors, including temperature, water activity, light, oxygen, pH, and ascorbic acid [6,7]. The effect of temperature on the color degradation of fruit leather during storage has been commonly reported [8,9,10,11]. Nevertheless, other important factors during the storage of fruit leathers, e.g., the lighting and relative humidity (RH), are still scarce in the literature.

Light exposure may cause adverse effects on foods, such as the oxidation of vitamins, lipids, and natural pigments, resulting in the loss of nutrients, formation of off-flavors, and discoloration. The decrease in the quality of photosensitive compounds of foods due to exposure to light depends on the intensity and spectrum of the light source, exposure time, and packaging material [12]. For attracting consumers, many foods, including photosensitive foods, such as fruit leathers, are stored in transparent packaging and displayed on highly illuminated shelves [13]. The storage RH is another critical parameter that influences fruit leather quality. Apple leather color changed as RH was increased from 11% to 65%, increasing darkening because of the nonenzymatic browning intensification [14]. Strawberry leather at RH below 33% at 25 °C keeps the moisture content below the monolayer moisture, which is an important indicator for the physicochemical stability of dried product storage [15]. Below the monolayer moisture content value, the rates of deteriorative reactions are minimal (except oxidation) [16].

Survival analysis is a methodology that has been applied to estimate the sensory shelf life of a wide range of food products [17,18,19,20,21]. This method is based on consumers’ acceptance or rejection of a stored food product, and its shelf life is estimated as the time needed to achieve a predetermined consumer rejection percentage [22]. For storing and analyzing a product, the current-status survival analysis methodology developed by Araneda et al. [23] can be used during sensory shelf-life study. Each consumer evaluates only one sample corresponding to one storage time. A minimum of 50 consumers is recommended to evaluate the shelf life for each of the six different storage times [24,25].

In this context, this study aimed to determine the color-based shelf life of strawberry leather during storage under a combination of different illumination and RH conditions. For this purpose, (i) changes in the instrumental color parameters and total anthocyanin content of the sample during storage were evaluated; (ii) current-status survival analysis was applied to estimate the sensory shelf life of the product; and (iii) the analytical attributes and consumer-rejection percentage of the strawberry leather were correlated.

## 2. Materials and Methods

### 2.1. Preparation of Strawberry Leather

Strawberries were purchased from the local market (Florianópolis, Brazil). The selection of fruits was based on their degree of ripeness, evaluated by visual analysis (the most reddish strawberries) and soluble solids concentration (6–8 Brix), determined with a refractometer (Atago, PAL-BX/RI, Tokyo, Japan). The pulp was prepared from selected strawberries that were washed and crushed in a household blender (Philco, São Paulo, Brazil), discarding the sepal and pedicel. The pulp was frozen at −18 °C and thawed according to the amount required for each test.

Strawberry leathers were produced by cast-tape drying equipment (CTD), operating in batch. During the drying process, the strawberry pulp was uniformly spread on a fiberglass support coated with Teflon^®^ (Lençol Armalon^®^ Standard, Indaco, São Paulo, Brazil) using a doctor blade with a 2 mm gap. The bottom face of the Teflon-coated film was heated by steam produced from hot water at 98 °C. An exhaustion/ventilation tunnel removed the evaporated water from the product during drying at an average air velocity of 1.2 m.s^−1^. The relative humidity of the ambient ranged from 54% to 79%, and the temperature was between 21 and 26 °C. The strawberry pulp was dried until the moisture content and water activity reached approximately 3 g/100 g (dry basis) and 0.350, respectively. The total drying time was 12 min.

### 2.2. Storage Conditions

Strawberry leathers were cut into 40 mm x 40 mm squares and equilibrated at 25 °C, in chambers with two saturated salts solutions, CH_3_COOK and Mg(NO_3_)_2_, giving RH of 22.5% and 52.3%, respectively. These conditions were chosen based on the critical RH value (33% RH) reported by Frabetti et al. [15] for strawberry leathers. At each one of these RHs, samples were stored with no illumination (black chamber) and with a cool white light-emitting diode (LED) lamp (9 W, color temperature of 6500 K) at 1010 ± 190 lx, which is close to that used in markets [26,27]. The spectrum of the cool white LED is presented in the study by Kim et al. [28]. This light source exhibits blue color as the dominant color, with a peak wavelength near 455 nm [29]. Estimation of the illuminance that was received by the leathers in a chamber (in lux) was performed with a luximeter (Minipa, MLM-1011, São Paulo, Brazil). The four experimental conditions were analyzed in a total of 7 storage times: 0, 7, 14, 28, 48, 70, and 90 days for samples conditioned at 22.5% RH, with and without illumination; and 0, 1, 1 ¾ (42 h), 2, 7, 14, and 21 days for samples conditioned at 52.3% RH, with and without illumination. The storage time of strawberry leathers equilibrated at 22.5% RH was extended in order to prove the positive effect of low RH on the product color. Before each analysis, a fresh sample was produced and used as a control sample.

### 2.3. Instrumental Color Measurement

The color parameters (*L**, *a**, and *b**) of the strawberry leathers during the storage were assessed by a computer vision system-CVS [30]. Images were obtained with a photographic camera (Nikon Corporation, Nikon D5500, Tokyo, Japan) and processed using ImageJ v. 1.6.0 software (National Institutes of Health, Bethesda, MD, USA). The conversion from the RGB system to the CIELab scale was performed using a color space converter plug-in. The total color difference (Δ*E**) was calculated according to Equation (1), using the fresh sample as a reference (*L_o_**, *a_o_**, and *b_o_**). The color parameters of the fresh sample were the average values from samples produced at each storage time, and it was performed in triplicate:(1)$$∆E^{*}=\sqrt{{\left(L^{*}-L_{o}^{*}\right)}^{2}+{\left(a^{*}-a_{o}^{*}\right)}^{2}+{\left(b^{*}-b_{o}^{*}\right)}^{2}}$$

### 2.4. Total Anthocyanin Content

The total anthocyanin content of the strawberry leather samples was quantified using the pH differential method proposed by Giusti and Wrolstad [31]. To obtain extracts, 0.6 g of sample were extracted three times with 25 mL of methanol/water/formic acid (50:48.5:1.5) in a sonication water bath (Unique, 1400A, Indaiatuba, Brazil) for 2 min, until the extinction of color. After each extraction, the extracts were centrifuged at 3400 rpm for 5 min, and the supernatants were filtered, pooled, and diluted to a final volume of 100 mL with distilled water. The results were expressed in mg of pelargonidin-3-glucoside/100 g dry extract, using a molar absorptivity (*ε*) of 15,600 L.mol^−1^.cm^−1^ and molecular weight of 433.2 g.mol^−1^. Absorbances were read at 496 and 700 nm, with pelargonidin-3-glucoside being the predominant anthocyanin in strawberries [32]. The total anthocyanin content was performed in triplicate throughout storage.

### 2.5. Kinetic Modeling of Color and Total Anthocyanin

Zero-, first-, second-order, and fractional conversion models were trialed to predict the kinetics of color and anthocyanin degradation during storage. The first-order fractional conversion model (Equation (2)) was selected as the best fit (based on the coefficient of determination, *R^2^*) to mathematically describe the experimental data kinetics of the instrumental color and total anthocyanin content. In the equation, *X* is the attribute value at a storage time *t* (days), *X_o_* is the attribute value at the zero-storage time, *k* is the apparent reaction rate constant (days^−1^), and *X_eq_* is the nonzero equilibrium value of the attribute at infinite storage. The parameters of the model were predicted by nonlinear regression using MATLAB^®^ software (MathWorks Inc., R2018b, Natick, MA, USA):(2)$$X=X_{eq}+\left(X_{o}-X_{eq}\right)exp\left(-k.t\right)$$

### 2.6. Sensory Analysis

For each storage time, 50–60 consumers that did not reject the fresh sample and had normal color vision, as evaluated by the Ishihara test for color blindness [33], were recruited from students and workers from the Federal University of Santa Catarina (UFSC, Florianópolis, Brazil). They were invited to evaluate the color of samples stored in the four different conditions by an acceptability test, using a 9-point structured hedonic scale (1 = dislike extremely to 9 = like extremely). The samples were presented monadically in random order. Furthermore, for each sample, consumers answered “yes” or “no” to the followed question: “Would you normally purchase this product considering its color?”.

This study was conducted in accordance with the Declaration of Helsinki and approved by the Human Research Ethics Committee of the Federal University of Santa Catarina, Brazil (protocol number: 2.241.837). Prior to the test sessions, all participants signed an informed consent form.

### 2.7. Statistical Analysis

One-way analysis of variance (ANOVA) was applied to analyze the instrumental color, total anthocyanin content, and acceptability scoring data with subsequent comparison of mean values by Tukey’s test at the 95% confidence level (*p* < 0.05), using the software Statistica 10.0 (StatSoft, Tulsa, OK, USA).

The physicochemical properties and the consumer rejection probability of the leathers were correlated by linear regression analysis using Statistica 10.0 (StatSoft, Tulsa, OK, USA), and the fit goodness was determined by the coefficient of determination (*R^2^*), coefficient of correlation (*r*), and *p*-value (*p*).

Survival analysis methodology was applied to estimate the color-based shelf life of the strawberry leathers using the consumer acceptance/rejection test results. Considering the random variable *T* as the storage time at which the consumer rejects a sample, the rejection function *F(t)* can be defined as the probability of a consumers’ rejection of the sample with a storage time lower than time *t*, i.e., *F*(*t*) = *P*(*T* ≤ *t*). A parametric model can describe the rejection function, which provides precise estimates of the rejection function. Since the rejection times are not normally distributed, the Weibull distribution was chosen for modeling the rejection function (Equation (3)) [34]:(3)$$F\left(t\right)=1-exp\left[-exp\left(\frac{ln\;\left(t\right)-\mu }{\sigma }\right)\right]$$
where *µ* (location parameter) and *σ* (shape parameter) are the models’ parameters. The Weibull distribution is a commonly selected distribution used to model consumers’ rejection due to its simplicity, flexibility, and great fit to data [35]. MATLAB^®^ software (MathWorks Inc., R2018b, Natick, MA, USA) was used to estimate the model parameters, considering a 95% confidence interval (CI). The shelf-life prediction was obtained considering 25% and 50% consumer rejection using the estimated parameters [22].

## 3. Results and Discussion

### 3.1. Instrumental Color Parameters

Color is an essential sensory attribute that determines food products’ acceptance or rejection by consumers. Figure 1 shows photographs of the strawberry leather samples conditioned at the different relative humidity (22.5% and 52.3%) and with and without LED illumination (0 and 1010 lx, respectively) during storage. Strawberry leather changed the color visually during storage, being mainly impacted by RH. For example, samples with a water activity of 0.523 (i.e., conditioned at 52.3% RH) changed from bright red on the day of their production to brown on day 21 of storage.

The instrumental color parameters (*L**, *a**, and *b**) determined using CIELab color space and total color difference (Δ*E**) of the strawberry leathers are shown in Figure 2. The *L**, *a**, and *b** values of the fresh sample (before storage) were 56.41 ± 2.38, 37.81 ± 2.13, and 21.85 ± 2.17, respectively. Abonyi et al. [36] reported similar results for the strawberry leathers obtained by Refractance Window drying. The *L** and *b** values remained constant during storage regardless of the RH and illumination, as shown in Figure 2(Ia,b,IIIa,b). As these parameters were not affected by the storage conditions (*p* > 0.05), the kinetic model was not used to describe them. On the other hand, the *a** value (redness) decreased throughout the storage time for all studied conditions (*p* < 0.05). Buvé et al. [3], Garzón and Wrolstad [5], and Agudelo-Laverde et al. [37] reported similar behavior for these color parameters of strawberry products (strawberry juice and dehydrated strawberry) during storage. The *a** value of the strawberry leather samples decreased faster during storage at the higher RH. Samples before storage showed an *a** value of 37.81 ± 2.13, and after 90 days at 22.5% RH, this value decreased to 30.93 ± 1.76 and 31.08 ± 1.08, without and with illumination, respectively, while samples conditioned at 52.3% RH after 21 days showed an *a** value of 22.74 ± 0.56 and 17.26 ± 0.67, without and with illumination, respectively. This result could be related to the anthocyanin degradation and non-enzymatic browning reactions, which are accelerated at higher water activities [37]. In general, the LED illumination did not influence the *a** values of the strawberry leather samples during storage at 22.5% RH (*p* > 0.05). However, samples conditioned at 52.3% RH with illumination presented a higher reduction of the *a** values than those conditioned with no illumination.

The Δ*E** values were mainly affected by RH, and the changes in this parameter were due to the decrease in the *a** value, while the *L** and *b** parameters remained constant during storage. Cserhalmi et al. [38] reported that the color change could be perceived by consumers depending on the Δ*E** value. The color difference is noticeable by consumers when the Δ*E** value is higher than 1.5. The strawberry leather samples quickly reached an Δ*E** value of 1.5 in all storage conditions, as shown in Figure 2(IVa,b). However, consumers may accept the samples even if color changes are visible [3].

The first-order fractional conversion model satisfactorily described the experimental results of the changes in the *a** and Δ*E** values of the strawberry leather during storage in different RH and illumination conditions (*R^2^* > 0.70). Other studies have also predicted the color alteration in foods during storage using the first-order fractional conversion model [3,39,40]. The estimated kinetic parameters are displayed in Table 1. For both color parameters, the estimated *k* value was higher for samples conditioned at 52.3% RH than those conditioned at 22.5% RH, indicating greater color change for the higher storage RH.

### 3.2. Total Anthocyanin Content

Anthocyanin pigments are relatively unstable and susceptible to degradation during storage, resulting in color changes of food products. Anthocyanin degradation is due to non-enzymatic oxidation reactions (direct oxidative mechanism and/or the oxygen oxidizes other compounds that react with anthocyanins), enzymatic reactions, and/or condensation reactions with other compounds [3]. Figure 3 presents the effect of the different storage conditions on the total anthocyanin content of the strawberry leather.

The total anthocyanin content significantly decreased throughout time for all storage conditions (*p* < 0.05), and its decreasing rate was mainly dependent on RH. The increase of RH led to higher anthocyanin degradation in both illumination conditions. Anthocyanin losses in samples equilibrated at 22.5% RH, without and with illumination, were approximately 37%, while losses for samples conditioned at 52.3% RH, without and with illumination, were 65% and 94%, respectively. Agudelo-Laverde et al. [37] and Syamaladevi et al. [41] also reported the negative effect of RH during storage on anthocyanin retention in dehydrated strawberry and raspberry. Higher water activity values result in a higher conversion from anthocyanins to a hydrated carbinol base, a less stable structure [42], explaining the highest anthocyanin retention of strawberry leather samples conditioned at 22.5% RH. There were no significant differences between the anthocyanin content of samples conditioned at 22.5% RH with no illumination and those stored with illumination (*p* > 0.05). On the other hand, light significantly affected anthocyanin retention in the strawberry leather samples equilibrated at 52.3% RH (*p* < 0.05). According to Wrolstad et al. [7], the presence of light will promote anthocyanin degradation, while low water activity will improve anthocyanin stability.

Most previous studies used first-order kinetics to describe anthocyanin degradation during storage. However, in this study, the change in the total anthocyanin content during storage was best described by the first-order fractional conversion model (Table 2). This model considers that the concentration of a chemical compound remains after the end of kinetic study under the experimental conditions (tailing effect). This behavior can be attributed to the amount of anthocyanin that remained in a bound form [43]. Tiwari et al. [43,44] reported a similar model for anthocyanins degradation in strawberry and grape juices during ozone processing.

Red anthocyanin pigment degradation during storage results in a reduction of the samples’ redness [3]. Thus, the total anthocyanin content was correlated to the *a** value to analyze the contribution of anthocyanin degradation to the color changes of the strawberry leather during storage. A linear equation was fitted to the experimental data of the *a** value versus the total anthocyanin content (Figure 4). The total anthocyanin content strongly correlated with the *a** value in the strawberry leather samples equilibrated at the higher RH (*R^2^* > 0.91). This result suggests that the *a** values can predict anthocyanin degradation at 52.3% RH. Agudelo-Laverde et al. [37] reported that red coordinate values of freeze-dried strawberries correlated with anthocyanin degradation at relative humidity above 43%. At 52.3% RH, the rate of the *a** value change in the function of the total anthocyanins content was higher for samples stored under illumination than those conditioned with no illumination. This could be explained by the presence of light in this relative humidity, which accelerates the degradation of anthocyanins and, consequently, the reduction of redness.

### 3.3. Sensory-Based Estimation of Shelf Life

The instrumental measures by themselves do not indicate the acceptability or rejection of the product by consumers. Therefore, the color-based acceptance of the strawberry leather during storage in the different RH and illumination conditions was evaluated by consumers (Figure 5). The consumer acceptability decreased with increasing storage time, showing a relation between the product’s acceptability and color. Buvé et al. [17] and Gössinger et al. [45] reported that color is a critical quality parameter in consumers’ acceptance of strawberry-based products. The reduction rate of the color-based acceptance of the strawberry leather samples was mainly dependent on the storage RH. Samples equilibrated at 52.3% RH showed a faster acceptability decrease during storage than samples conditioned at 22.5% RH. On the other hand, the acceptability of the strawberry leather samples was not significantly affected by LED illumination during storage in both relative humidity, even though this factor influenced the *a** and Δ*E** values at 52.3% RH.

In the survival analysis, the parametric Weibull model was used to predict the color-based shelf life of the strawberry leather samples. The parameters (*µ* and *σ*) of the Weibull distribution and their respective standard errors for each storage condition are presented in Table 3. Using these parameters, the consumer rejection percentage curves as a function of the storage time were plotted in Figure 6. From these curves, the shelf life was estimated for each storage condition, considering the 25% and 50% rejection probability of the product by consumers (Table 4). These results evidence the accelerating effect of storage RH on the color-based rejection probability and product shelf life. The increase of the RH resulted in a faster increase of the consumer rejection probability and a shorter shelf life of the strawberry leather samples. For example, for a 50% consumer rejection probability, the estimated sensory shelf life of the strawberry leather samples conditioned at 22.5% RH was at least 54 days, while it was reduced to approximately 2 days at 52.3% RH, regardless of the illumination condition. This result demonstrates the importance of RH’s control during storage to preserve the color and extend the sensory shelf life of the strawberry leather. Packaging with a high barrier for water vapor and a low barrier for LED light can be a suitable alternative to achieve this goal.

### 3.4. Correlation between Physicochemical and Sensory Measurements

Although the consumers’ acceptance/rejection test for shelf life prediction of food is the most suitable tool, it is expensive, time-consuming, and difficult to perform routinely by food companies. Thereby, it would be more effortless for food industries to have an instrumental attribute correlated with consumer acceptability, which can be used as an indicator of the shelf life prediction [17,21]. This study carried out a linear correlation analysis between the strawberry leather consumer rejection probability and physicochemical properties (Table 5). The *a** value was negatively correlated with the percentage of consumer rejection for all studied storage conditions (*r* > −0.90). Therefore, the *a** value can be used as an indicator of the shelf life.

A linear equation was fitted to the experimental data of the *a** value versus the rejection consumer percentage for each storage condition (Figure 7). The cut-off values, which correspond to a 25% and 50% consumer rejection, were determined from this equation. A slight variation in the *a** value caused a significant increase in the samples’ rejection equilibrated at both relative humidity. Strawberry leathers conditioned at 22.5% RH showed a rejection of 25% with *a** values of around 34.5, while for 50% of rejection, the *a** value was around 32.0 (with and without LED illumination). Samples conditioned at 52.3% RH would be rejected by more than 25% of consumers when the *a** values were around 34.0 (with and without LED illumination), while 50% of the consumers rejected the samples when *a** values decreased to 30.48 and 28.45, without and with LED illumination, respectively. These limit values could be used as a fast tool for shelf life predictions of strawberry leather.

## 4. Conclusions

The results of this study indicated that the product’s water activity plays a crucial role in color change during storage. In strawberry leather samples equilibrated at the higher relative humidity, the decrease of the anthocyanin content strongly correlated with the decrease of the red chromatic parameter (*a** value), suggesting that *a** values can be used as an index of anthocyanin degradation in strawberry leathers. Current-status survival analysis was a suitable methodology to estimate the sensory shelf life of strawberry leather. The percentage of consumer rejection of the color analysis was highly correlated with the *a** value, which may be used as an analytical indicator for estimating strawberry leather’s shelf life. Based on this result, mathematical functions predicting the evolution of the *a** value as a function of the consumer rejection probability of the product were developed. The approach followed in this research, in which instrumental and sensorial data were integrated to evaluate color changes during storage and to predict the product’s shelf life in the market, is a useful tool for application in industry quality-control programs.

## Figures and Tables

**Figure 1 foods-11-00218-f001:**
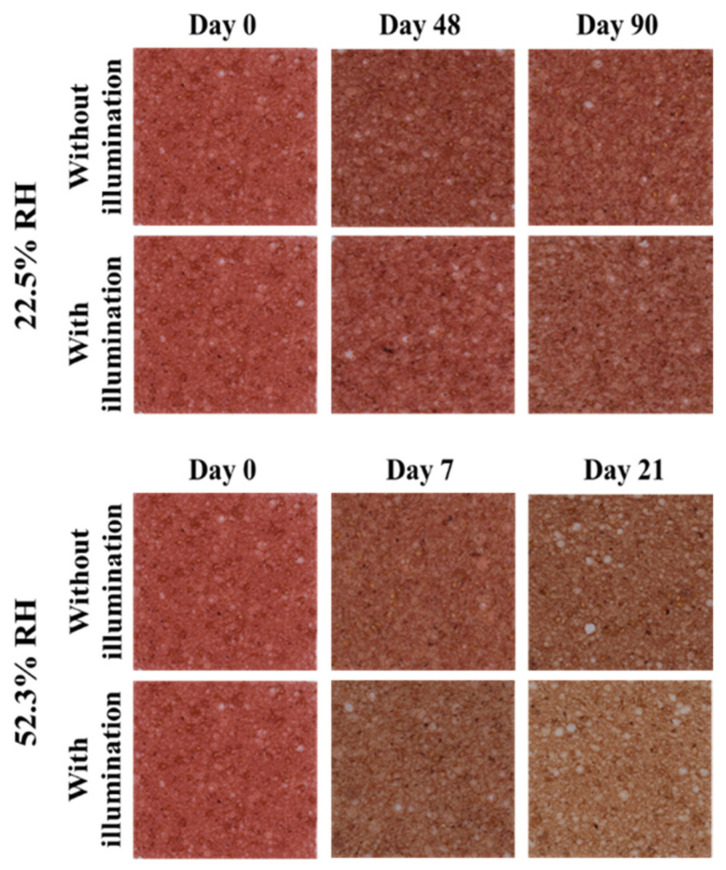
Evolution of color of the strawberry leather during storage.

**Figure 2 foods-11-00218-f002:**
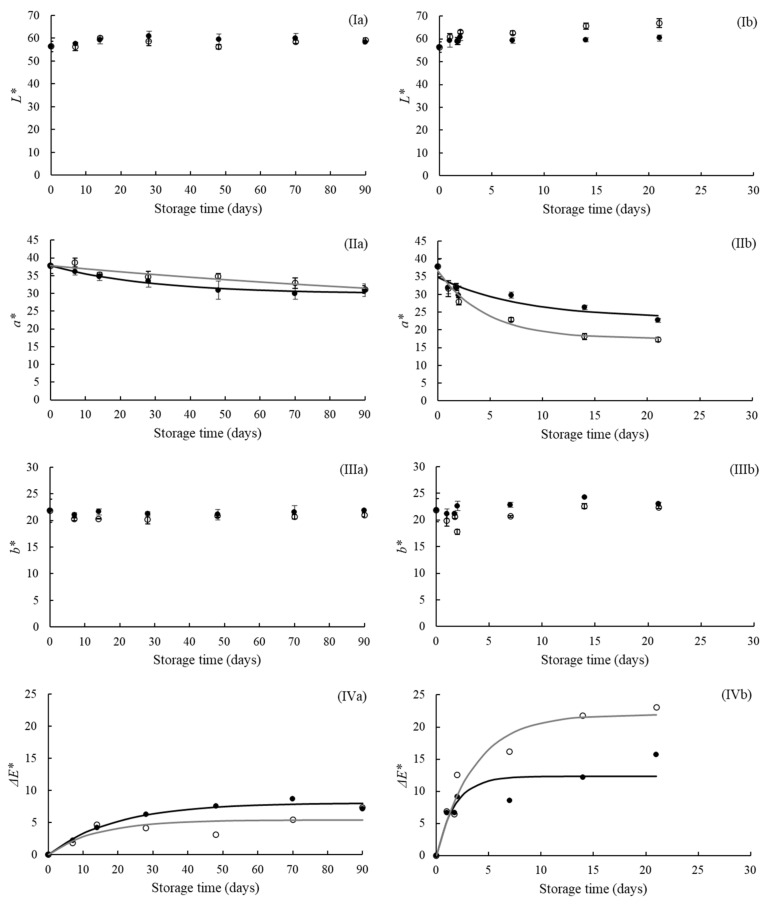
Evolution of color parameters (*L** (**I**), *a** (**II**), and *b** (**III**)) and total color difference (Δ*E** (**IV**)) of the strawberry leather during storage at 22.5% RH (**a**) and 52.3% RH (**b**), with no illumination (●) and with illumination (○). The filled and empty symbols represent experimental data. The full lines represent the first-order fractional conversion model.

**Figure 3 foods-11-00218-f003:**
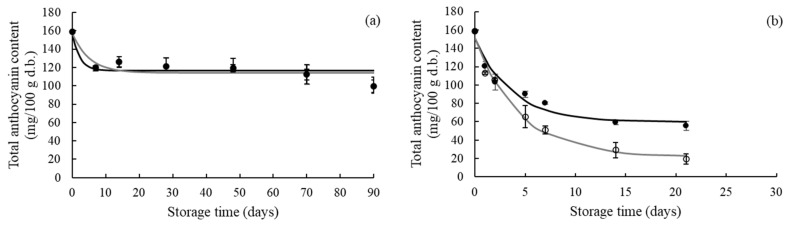
Evolution of the total anthocyanin content of strawberry leather during storage at 22.5% RH (**a**) and 52.3% RH (**b**), with no illumination (●) and with illumination (○). The filled and empty symbols represent experimental data. The full lines represent the first-order fractional conversion model.

**Figure 4 foods-11-00218-f004:**
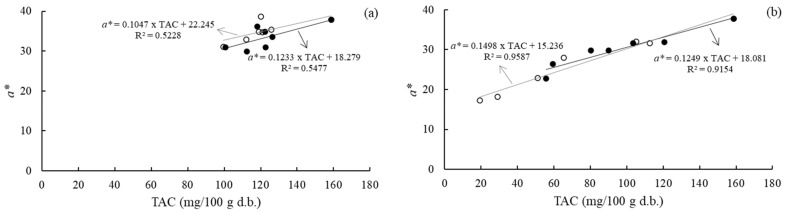
Relationship between the total anthocyanin content (TAC) and *a** chromatic parameter of strawberry leather during storage at 22.5% RH (**a**) and 52.3% RH (**b**), with no illumination (●) and with illumination (○).

**Figure 5 foods-11-00218-f005:**
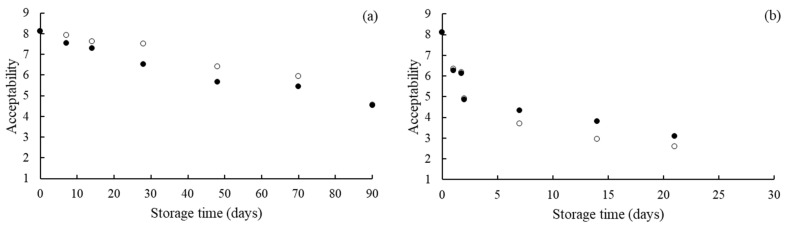
Consumers’ acceptability of strawberry leather during storage at 22.5% RH (**a**) and 52.3% RH (**b**), with no illumination (●) and with illumination (○).

**Figure 6 foods-11-00218-f006:**
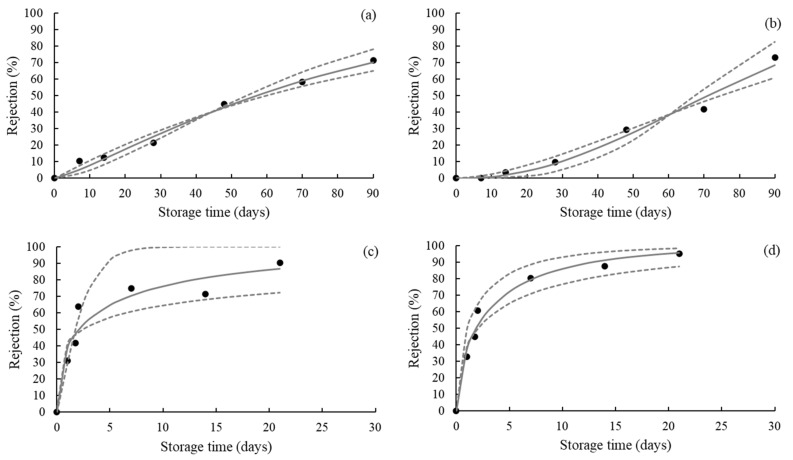
Consumer rejection probability of strawberry leather during storage at 22.5% RH with no illumination (**a**), 22.5% RH with illumination (**b**), 52.3% RH with no illumination (**c**), and 52.3% RH with illumination (**d**). The filled symbols represent experimental data. The full lines represent the parametric rejection function based on the Weibull distribution. The dashed lines are confidence intervals for 95%.

**Figure 7 foods-11-00218-f007:**
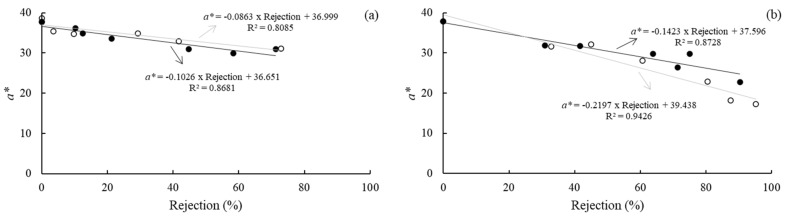
Relationship between the consumer rejection probability and *a** chromatic parameter of the strawberry leather during storage at 22.5% RH (**a**) and 52.3% RH (**b**), with no illumination (●) and with illumination (○).

**Table 1 foods-11-00218-t001:** Kinetic parameters for changes in *a** and Δ*E** values during storage of the strawberry leather.

Sample	*a**				Δ*E**			
	*X_0_*	*X_eq_*	*k* (Days^−1^)	*R^2^*	*X_0_*	*X_eq_*	*k* (Days^−1^)	*R^2^*
22.5% RH/Without illumination	37.88	29.87	0.034	0.972	0	8.07	0.052	0.975
22.5% RH/With illumination	37.91	22.75	0.006	0.872	0	5.43	0.072	0.707
52.3% RH/Without illumination	34.79	23.15	0.126	0.819	0	12.34	0.572	0.812
52.3% RH/With illumination	36.77	17.44	0.219	0.970	0	21.98	0.279	0.943

**Table 2 foods-11-00218-t002:** Kinetic parameters for anthocyanins degradation during storage of strawberry leather.

Sample	Total Anthocyanin Content (mg/100 g d.b.)			
	*X_0_*	*X_eq_*	*k* (Days^−1^)	*R^2^*
22.5% RH/Without illumination	158.6	116.8	0.458	0.774
22.5% RH/With illumination	158.0	114.1	0.198	0.801
52.3% RH/Without illumination	150.5	59.6	0.274	0.957
52.3% RH/With illumination	151.1	21.9	0.227	0.985

**Table 3 foods-11-00218-t003:** Values of the model parameters (*µ* and *σ*) for each storage condition using a Weibull distribution for survival analysis.

Sample	*µ*	*σ*
22.5% RH/Without illumination	4.341 ± 0.112	0.821 ± 0.170
22.5% RH/With illumination	4.433 ± 0.105	0.465 ± 0.155
52.3% RH/Without illumination	1.515 ± 0.661	2.157 ± 1.337
52.3% RH/With illumination	1.218 ± 0.293	1.612 ± 0.511

**Table 4 foods-11-00218-t004:** Estimated sensory shelf life values of the strawberry leather during storage for 25% and 50% consumer rejection, and their 95% confidence intervals (CIs).

Sample	Shelf Life (Days)	
	For 25% Rejection (95% CI)	For 50% Rejection (95% CI)
22.5% RH/Without illumination	27.6 (25.0–30.5)	56.8 (54.1–59.7)
22.5% RH/With illumination	47.2 (43.2–51.5)	71.0 (67.7–74.5)
52.3% RH/Without illumination	0.3 (0.1–0.8)	2.1 (1.7–2.4)
52.3% RH/With illumination	0.5 (0.3–0.6)	1.9 (1.7–2.1)

**Table 5 foods-11-00218-t005:** Correlation coefficients (*r*) between the consumer rejection probability and physicochemical properties of strawberry leather during storage.

Physicochemical Parameter	Consumer Rejection			
	22.5% RH/ without Illumination	22.5% RH/ with Illumination	52.3% RH/ without Illumination	52.3% RH/ with Illumination
*L**	0.37	0.32	0.84 *	0.94 *
*a**	−0.93 *	−0.90 *	−0.93 *	−0.97 *
*b**	0.20	0.17	0.67	0.19
Δ*E**	0.82 *	0.80	0.86 *	0.97 *
Total anthocyanins content	−0.76 *	−0.73	−0.97 *	−0.99 *

* *p* < 0.05.

## Data Availability

Not applicable.

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
