# Peer review of "Survival Analysis to Predict How Color Influences the Shelf Life of Strawberry Leather"

_foods, 2022, doi:10.3390/foods11020218_

Round 1
Reviewer 1 Report
Figure 3 (b), Figure (Ib, IIb, IIb, and IVb), Figure 5 (b) and Figure 6 (b) the days must reduce to 30.
The conclusion. The line 367 to 370 must be erased. The results indicating that relative humidity plays a crucial role......
Author Response
Reviewer #1:
Figure 3 (b), Figure (Ib, IIb, IIb, and IVb), Figure 5 (b) and Figure 6 (b) the days must reduce to 30.
Answer: The Figures were improved in the new version of the manuscript as suggested.
The conclusion. The line 367 to 370 must be erased. The results indicating that relative humidity plays a crucial role......
Answer: The conclusion was improved as suggested.
Reviewer 2 Report
the experimental work is well structured. I suggest making the following changes:
lines 111-121: it would be useful to also enter the values ​​of C and H to have a better evaluation of the color, since it is also used as an evaluation parameter in the results section.
The authors should explain how the shelf life calculation was made, in particular the times indicated in table 4, given that in the materials and methods section only the modeling of color and total anthocyanins (paragraph 2.5) and the analysis are reported. sensory (paragraph 2.6), which do not allow to calculate the shelf life time.
Author Response
the experimental work is well structured. I suggest making the following changes:
lines 111-121: it would be useful to also enter the values of C and H to have a better evaluation of the color, since it is also used as an evaluation parameter in the results section.
Answer: b* value remained constant during the storage, and C and H values are dependent of b* and a*, so the changes are attributed to a* value. Therefore, we preferred to continue with the discussion using L*, a*, and b*, because we used the ΔE* value, which is also dependent on L*, a*, and b*. That is used as an indicator to check whether the color difference can be perceived by consumers (Cserhalmi et al., 2006).
Cserhalmi, Zs.; Sass-Kiss, Á.; Tóth-Markus, M.; Lechner, N. Study of pulsed electric field treated citrus juices. Innov. Food Sci. Emerg. Technol. 2006, 7, 49-54. https://doi.org/10.1016/j.ifset.2005.07.001.
The authors should explain how the shelf life calculation was made, in particular the times indicated in table 4, given that in the materials and methods section only the modeling of color and total anthocyanins (paragraph 2.5) and the analysis are reported. sensory (paragraph 2.6), which do not allow to calculate the shelf life time.
Answer: The shelf-life was estimated according to the Weibull model described in the topic 2.7 of “Materials and Methods” section. The time calculation indicated in Table 4 was explained in Lines 332-338 of the new version of the manuscript.
Reviewer 3 Report
This research tried to predict how the color of strawberry leather influences its shelf life, using a survival analysis. This reviewer thinks that purpose, methodology, results, and conclusion are reasonable. Especially, the Introduction seems to be well-written and clear. To increase the value of your study, please consider the following comments.
Abstract
This reviewer thinks that the background and purpose of this study should be mentioned briefly in this section.
Materials and Methods
2.2
Why did you choose 25 °C, and 22.5% and 52.3%?
Regarding the light condition, the information about distribution of the wavelength is necessary. To show as a figure is preferred.
Results and Discussion
It seems that the statistical results based on Tukey’s test are missing from each figure or table.
Table 5
P < 0.05: Is this based on what kind of statistical method?
Author Response
Reviewer #3:
This research tried to predict how the color of strawberry leather influences its shelf life, using a survival analysis. This reviewer thinks that purpose, methodology, results, and conclusion are reasonable. Especially, the Introduction seems to be well-written and clear. To increase the value of your study, please consider the following comments.
Abstract
This reviewer thinks that the background and purpose of this study should be mentioned briefly in this section.
Answer: The Abstract was improved as suggested.
Materials and Methods
2.2 Why did you choose 25 °C, and 22.5% and 52.3%?
Answer: Fruit leather is a product that does not require refrigeration during storage, that why we choose the room temperature (25 °C). The storage of the strawberry leather in relative humidities of 22.5% and 52.3% result in microbiological stable products, according to Labuza (1980). Also, in our previous study Frabetti et al. (2021) strawberry leathers should be kept stored at RH lower than 33% to below the monolayer value, which is the minimum moisture content strongly adsorbed on the hydrophilic sites of the product's surface. It is also a measure of the availability of these sites for water sorption on the material, besides being one of the indicators of the optimal storage conditions used to control products' stability. So, we choose to work with a higher and lower RH than this critical value.
Labuza, T. P. (1980). The effect of water activity on reaction kinetics of food deterioration. Food Technology, 34(59), 36–41.
Frabetti, A.C.C., de Moraes, J.O., Porto, A.S., Simão, R.S., Laurindo, J.B. (2021). Strawberry-hydrocolloids dried by continuous cast-tape drying to produce leather and powder, Food Hydrocolloids, 121,107041, https://doi.org/10.1016/j.foodhyd.2021.107041.
Regarding the light condition, the information about distribution of the wavelength is necessary. To show as a figure is preferred.
Answer: The following text was added in Lines 117-119: “The spectrum of the cool white LED is presented in the study by Kim et al. [23]. This light source exhibits blue color as the dominant color, having the peak wavelength near 455 nm [24].”
Results and Discussion
It seems that the statistical results based on Tukey’s test are missing from each figure or table.
Answer: The statistical analysis based on ANOVA and Tukey’s test was performed using results from the instrumental color parameters, total anthocyanin content, and acceptability scoring. These experimental data were shown in the manuscript in form of graphs. Thus, for better visualization of these figures, the symbols that represent whether there is a statistical difference were not presented. However, the discussion of results in the manuscript text informed the statistical results.
Table 5
P < 0.05: Is this based on what kind of statistical method?
Answer: The consumer rejection and physicochemical properties were correlated (Table 5) by linear regression analysis using Statistica, and the goodness of fitting was evaluated based on statistical parameters of fitting (R², r, and p). Statistical method was described in Lines 185-188 of the new version of the manuscript.